# In Vitro Anticancer Activity and Oxidative Stress Biomarkers Status Determined by *Usnea barbata* (L.) F.H. Wigg. Dry Extracts

**DOI:** 10.3390/antiox10071141

**Published:** 2021-07-20

**Authors:** Violeta Popovici, Laura Bucur, Gabriela Vochita, Daniela Gherghel, Cosmin Teodor Mihai, Dan Rambu, Suzana Ioana Calcan, Teodor Costache, Iulia Elena Cucolea, Elena Matei, Florin Ciprian Badea, Aureliana Caraiane, Victoria Badea

**Affiliations:** 1Department of Microbiology and Immunology, Faculty of Dental Medicine, Ovidius University of Constanta, 7 Ilarie Voronca Street, 900684 Constanta, Romania; violeta.popovici@365.univ-ovidius.ro (V.P.); victoria.badea@365.univ-ovidius.ro (V.B.); 2Department of Pharmacognosy, Faculty of Pharmacy, Ovidius University of Constanta, 6 Capitan Al. Serbanescu Street, 900001 Constanta, Romania; laurabucur@univ-ovidius.ro; 3NIRDBS, Institute of Biological Research Iasi, 47 Lascar Catargi Street, 700107 Iasi, Romania; mihai.cosmin.teo@gmail.com; 4Advanced Centre for Research and Development in Experimental Medicine (CEMEX), “Grigore T. Popa” University of Medicine and Pharmacy Iasi, 9–13 Mihail Kogalniceanu Street, 700259 Iasi, Romania; 5Research Center for Instrumental Analysis SCIENT, 1E Petre Ispirescu Street, 077167 Tancabesti, Ilfov, Romania; dan.rambu@scient.ro (D.R.); suzana.calcan@scient.ro (S.I.C.); teodor.costache@scient.ro (T.C.); iulia.cucolea@scient.ro (I.E.C.); 6Center for Research and Development of the Morphological and Genetic Studies of Malignant Pathology, Ovidius University of Constanta, CEDMOG, 145 Tomis Blvd., 900591 Constanta, Romania; sogorescuelena@gmail.com; 7Department of Oral Rehabilitation, Faculty of Dental Medicine, Ovidius University of Constanta, 7 Ilarie Voronca Street, 900684 Constanta, Romania; florin.badea@365.univ-ovidius.ro (F.C.B.); aureliana.caraiane@365.univ-ovidius.ro (A.C.)

**Keywords:** *Usnea barbata* dry extracts, usnic acid, CAL-27 cancer cells, V79 healthy cells, cytotoxicity, clonogenesis, wound healing assay, antioxidant enzymes activity

## Abstract

Lichens represent an important resource for common traditional medicines due to their numerous metabolites that can exert diverse pharmacological activities including anticancer effects. To find new anticancer compounds with fewer side effects and low tumor resistance, a bioprospective study of *Usnea barbata* (L.) F.H. Wigg. (*U. barbata*), a lichen from the Călimani Mountains (Suceava county, Romania) was performed. The aim of this research was to investigate the anticancer potential, morphologic changes, wound healing property, clonogenesis, and oxidative stress biomarker status of four extracts of *U. barbata* in different solvents (methanol, ethanol, acetone, and ethyl acetate), and also of usnic acid (UA) as a positive control on the CAL-27 (ATCC^®^ CRL-2095™) oral squamous carcinoma (OSCC) cell line and V79 (ATCC^®^ CCL-93™) lung fibroblasts as normal cells. Using the MTT assay and according to IC_50_ values, it was found that the most potent anticancer property was displayed by acetone and ethyl acetate extracts. All *U. barbata* extracts determined morphological modifications (losing adhesion capacity, membrane shrinkage, formation of abnormal cellular wrinkles, and vacuolization) with higher intensity in tumor cells than in normal ones. The most intense anti-migration effect was established in the acetone extract treatment. The clonogenic assay showed that some *U. barbata* extracts decreased the ability of cancer cells to form colonies compared to untreated cells, suggesting a potential anti-tumorigenic property of the tested extracts. Therefore, all the *U. barbata* extracts manifest anticancer activity of different intensity, based, at least partially, on an imbalance in antioxidant defense mechanisms, causing oxidative stress.

## 1. Introduction

Nowadays, cancer still remains one of the most important causes of death, representing a challenge for the scientific community to search for new bioactive molecules that are effective and safe in the fight against this implacable disease of the contemporary world [1,2]. Particularly, OSCC represents more than 90% of all oral neoplasms, which have as common sites of development the tongue, lips, and floor of the mouth. This type of neoplasm has a high mortality rate (5-year survival rate of around 50%) because numerous cases are diagnosed at a late stage of the disease [3]. The major purpose of current cancer therapy is to perturb or kill the malignant cells, from early stages neoplasms, or those remaining after surgery, without affecting the normal cytophysiological processes, and to trigger the immunity defense against the residual tumor cells. Most chemotherapeutic agents used in cancer treatment are toxic at effective doses, affecting the malignant cells and the normal ones from the body, require high dose therapy, have adverse side effects, develop multiple drug resistance (MDR) of cells, and are immunosuppressive. Therefore, finding new pharmacological agents with anticancer activity constitutes an essential concern of oncological research and medical practice to improve antitumor therapy effectiveness.

Natural products represent rich sources of bioactive molecules for new anticancer drugs [4,5], and phytochemicals show great potential in improving the clinical condition in cancer patients, as demonstrated by numerous preclinical and clinical studies. Moreover, plant extracts and their secondary metabolites could promote cancer cell death by various molecular mechanisms [6,7,8,9]. Therefore, programmed tumor cell death (PCD) is one of the most known anticancer mechanisms and has three forms: apoptosis, necroptosis, and pyroptosis [10]. All of these PCDs vary directly proportional to ROS levels [11,12,13]. Apoptosis represents one of the most studied processes, consisting of non-inflammatory PGD. Numerous plant extracts and natural compounds induce apoptosis in neoplastic cells [14,15,16]. Furthermore, pyroptosis is an emerging inflammatory cell death mode, precepted as secondary necrosis after apoptosis; it is a lytic process that can be induced by the progressive loss of plasma membrane integrity of apoptotic cells [17,18]. Both PCD processes are caspase-dependent molecular mechanisms [19]; in addition, Jiang et al. (2020) highlighted that the caspase-3/GSDME signal pathway is a switch between apoptosis and pyroptosis in cancer [17]. Finally, necroptosis is an alternative mode of pro-inflammatory PCD, surnamed programmed necrosis, overcoming apoptosis. It also reportedly serves as a “fail-safe” mechanism that protects against tumor development when resistance processes compromise apoptosis [20,21]. With ferroptosis and parthanatos, necroptosis represents a caspase-independent regulated necrosis pathway [22]. Necroptosis and pyroptosis trigger pro-inflammatory signals into the cellular surroundings; contrariwise, apoptosis dampens subsequent immune responses [10] (p. 1106). Several scientific studies have reported that natural products could induce pro-inflammatory programmed tumor cell death. For instance, *Oregano vulgare* L., [23], *Acridocarpus orientale* [24], and *Galenia africana* [25] registered antitumor effects by apoptosis and necroptosis. Pujari et al. (2021) proved that moscatylin (a plant-derived stilbenoid compound) displays anticancer effects by apoptosis and pyroptosis [26]. Another study proved that ferroptosis, apoptosis, and autophagy are antitumor mechanisms of *Thymus vulgaris* and *Arctium lappa* on leukemia and multiple myeloma tumor cell lines [27].

In the last years, novel cancer treatment strategies exploiting pro-senescence therapies have attracted remarkable interest [28,29]. Cellular senescence represents a state of stable cell cycle arrest that could be promoted as a response to various injuries; this state consists of indistinct morphological hallmarks, gene expression profiles, and the senescence-associated secretory phenotype (SASP) [30]. Cellular senescence is an important component of normal physiology with tumor-suppressive functions [31]. As a natural compound, hinokitiol (natural monoterpenoid) induces DNA damage and autophagy, followed by cell cycle arrest and senescence in gefitinib-resistant lung adenocarcinoma cells, as reported Li et al. (2014) [32]. In another study, Zheng et al. (2017) proved that an optimized plant mixed formula had anticancer action inducing cell cycle arrest and senescence against lung tumor cells [33].

However, cell cycle arrest cannot mean only senescence [34]. The cell cycle can be stopped in various phases (G0/G1; G1/G2; G2/M) and cell proliferation could be inhibited. Thus, Nguyen et al. (2017) communicated that quercetin promotes tumor cell apoptosis and cell cycle arrest in G2/M [35]. In other studies, *Litsea cubeba* (Lour.) heartwood and fruit extracts showed a cytotoxic effect against T47D breast cancer cells, arresting the cell cycle in G0/G1 [36]; likewise, De Souza Grinevicius et al. (2016) described *Piper nigrum* ethanolic extract anticancer activity, which causes high ROS production, oxidative damage in DNA inducing cell cycle arrest, and apoptosis in cancer cells [37].

A large part of biomolecules derived from plants can act synergistically with chemo- and radiotherapy. This association could increase the therapeutic effects and could reduce the side effects due to lower doses of conventional therapeutics needed [38,39,40,41,42,43]. Lichens are part of a particular group of symbiotic organisms, representing the association between a fungus and a green alga or a cyanobacterium, producing numerous secondary metabolites [44,45]. Lichens have been used in traditional medicine from ancient times [46]. It has been observed that many lichens have an unpleasant taste and can serve as a defense against herbivores. The secondary metabolites manifest significant biological and pharmacological properties such as antibacterial, antiviral, anti-inflammatory, antipyretic, anti-proliferative, and cytotoxic effects [47,48,49]. Moreover, these compounds have revealed antineoplastic activities in preclinical studies [50,51,52,53,54]. One of the secondary metabolites of lichens is usnic acid (C_18_H_16_O_7_) [2,6-diacetyl-7,9-dihydroxy-8,9b-dimethyldibenzofuran-1,3(2H,9bH)-dione)], predominantly found in the *Alectoria*, *Cladonia*, *Evernia*, *Lecanora*, *Ramalina*, and *Usnea* genera, constituting about 4–8% of the dry weight of thalli, which depends on the environmental conditions [55]. It has been shown that UA manifests antiviral, antimicrobial, antiprotozoal, anti-inflammatory, anti-metastatic, anti-angiogenic, anti-proliferative, wound-healing, and analgesic activity, making this compound an interesting subject for the pharmaceutical industry [56,57,58,59].

Therefore, our research aimed to investigate the anticancer potential of some extracts of *Usnea barbata* (L.) F.H. Wigg (named Song Luo in China, commonly called Old Man’s Beard), a lichen used for over 2000 years in Chinese traditional medicine [60]; this lichen was collected from the Călimani Mountains, Romania. Four dry extracts of *Usnea barbata* (L.), using various solvents, were obtained, in which the usnic acid, polyphenols, and tannins were previously quantified [61]. The in vitro antitumor property of these four dry extracts on the CAL-27 cancer cell line (tongue) was highlighted by assessing their cytotoxic activity, morphological changes, wound healing capacity, clonogenesis, and antioxidant enzyme system activity. Likewise, an evaluation of in vitro toxicity of the studied extracts was realized on non-cancerous V79 cells (lung fibroblasts). This study emphasizes the usefulness of the lichens in cancer treatment and their pharmacological potential, making them good candidates for new drug discovery.

## 2. Materials and Methods

### 2.1. Preparation and Characterization of Lichen Extracts

Four *U. barbata* extracts were studied using different solvents: methanol extract (UBM), ethanol extract (UBE), acetone extract (UBA), and ethyl acetate extract (UBEA). Additionally, as a positive control, UA was tested and obtained according to the protocol presented in our previous work [61] (p. 919). *U. barbata* was harvested from Calimani Mountains forests, Suceava County, Romania. The fresh lichen was cleaned of impurities and dried at 18–25 °C and sheltered from the sun in a herbal room. After drying, the obtained herbal product was preserved for a long time in the same conditions for subsequent studies. The lichen species identification was performed by the Department of Pharmaceutical Botany of the Faculty of Pharmacy, Ovidius University of Constanta, using standard methods. The dried lichen was ground to a powder; 60 g dried were extracted in a Soxhlet continuous reflux system, for eight hours with 450 mL of each solvent: acetone, ethanol (Chimreactiv S.R.L., Bucharest, Romania), ethyl acetate, methanol (Chemical Company, Iasi, Romania),) The extraction process was different for each extract, being around the boiling point of each solvent (Table 1). The rotary evaporator TURBOVAP 500 Caliper (Hopkinton, MA, USA) was used for evaporation of the solvents. Next, these extracts were kept for 16 h in a chemical exhaust hood for each optimal solvent evaporation. The obtained dry extracts were transferred to sealed-glass bottles and stored in the freezer (Sirge^®^, Avigliana (TO) Italia) at −24 °C until processing [61] (p. 919) (Appendix A).

The IR spectra of the four *U. barbata* dry extracts were obtained using the Frontier FTIR spectrometer produced by PerkinElmer^®^ (Waltham, MA, USA) equipped with the ATR (attenuated total reflectance) accessory in the range 4000–400 cm^−1^ at a resolution of 4 cm^−1^. Several mg of each sample was placed on the ATR crystal surface, and the obtained spectrum was the average of 16 readings. Each sample was analyzed in duplicate. The obtained results (with Spectrum™ 10 software from PerkinElmer^®^, Waltham, MA, USA) were interpreted based on the table with IR spectra according to the frequency range provided online by the Sigma Library of FTIR Spectra, Volumes 1 and 2 [62,63,64] and compared to those found in the literature.

### 2.2. Cell Lines and Cell Culture

The tumor and normal cell lines, respectively human OSCC cells CAL-27 (ATCC^®^ CRL-2095™) and normal lung fibroblasts from Chinese hamster V79 (ATCC^®^ CCL-93™), were obtained from American Type Culture Collection (ATCC, Rockville, MD, USA). Both cell lines were cultured in Dulbecco’s modified Eagle’s medium (DMEM, Biochrom AG, Berlin, Germany) with fetal bovine serum (FBS, Thermo Fisher Scientific, Berlin, Germany), 10%), antibiotic solution (Biochrom AG, Berlin, Germany) containing penicillin (100 IU/mL), and streptomycin (100 µg/mL), in a Binder incubator (BINDER GmbH, Tuttlingen, Germany) at 37 °C and 5% CO_2_ to guarantee growth and viability.

### 2.3. MTT Assay

MTT assay is a colorimetric method, modified after Mosmann (1983) [65] and Laville et al. (2004) [66], which is an exact and sensitive method suitable for adherent cell cultures, based on the capacity of NAD(P)H-dependent cellular oxidoreductase to convert yellow soluble tetrazolium [3-(4, 5-dimethyl thiazolyl-2)- 2,5-diphenyltetrazolium bromide] (MTT) into insoluble (E, Z)-5-(4,5-dimethylthiazol-2-yl)-1,3-diphenyl formazan (formazan) [67], subsequently dissolved with DMSO, resulting in a purple color whose intensity is directly proportional to the number of living cells [68], with the absorbance being converted in a number of cells based on standard curves realized with known cell dilutions [69]. The absorbance was measured at 570 nm with a Biochrom EZ Read 400 automatic microplate reader (Biochrom AG, Berlin, Germany). The cellular viability expressed as a percentage was calculated with the formula:Cell viability (%) = (Abs.) Test/(Abs.) Control × 100,
where Abs is the absorbance.

The *U. barbata* extracts were dissolved in DMSO 0.2%, then diluted with culture medium to realize different concentrations (from 25 to 200 µg/mL); usnic acid was also dissolved in DMSO 0.2% and diluted to obtain a range of doses from 1.25 to 10 µg/mL and used as a positive control. The untreated cells with the same volume of culture medium were the control group. The adherent cells were seeded in 96-well plates (6 × 10^3^ cells/well for V79 cells and 9 × 10^3^ cells/well for CAL-27 cells). After incubation for 24 h (monolayer formation), cells were treated 24 and 48 h with 25, 50, 100, and 200 µg/mL *U. barbata* extracts and 1.25, 2.5, 5, and 10 µg/mL usnic acid. Subsequently, the culture medium was replaced with 100 µL fresh medium, and 10 µL MTT (5 mg/mL) solution was added to each well and incubated for 3 h. Then, 90 µL of the mixture was removed, and 50 µL of DMSO was added into each well to dissolve the formazan crystals, the absorbance being measured with a microplate reader. The IC_50_ values were calculated from the concentrations of the *U. barbata* extracts that induced 50% inhibition of cell growth. Each test was repeated three times independently.

### 2.4. Cell Morphology Assay

The CAL-27 and V79 cells morphology was observed by the Nikon Eclipse TS 100 inverted microscope (Nikon, Tokyo, Japan) equipped with MshOt MS60 (Nikon, Tokyo, Japan) digital microscope camera after 24 and 48 h treatment of the *U. barbata* extracts. The images were saved as JPEGs.

### 2.5. In Vitro Wound Healing Assay

The cell migration was estimated by the in vitro scratch method [70,71]. The cells were seeded in 24 well plates, 3 × 10^4^ cells/well for V79 cells and 5 × 10^4^ cells/well for CAL 27 cells. After 24 h of monolayer formation, the scratch was made using a sterile 200 µL pipette tip, and the cells were washed twice with ice-cold phosphate-buffered saline (PBS) with pH 7.4 to remove the cell debris. Afterward, the cells were treated for 24 h with the concentration values of IC_50_ of the four *U. barbata* extracts and usnic acid established from the MTT assay for 48 h. The cells without treatment were used as the negative control and 0.2% DMSO as the vehicle control. The first set of images showing the edge of the scratch was observed immediately after incubation at 0 h (T0) with an inverted microscope (Nikon Eclipse TS 100, Tokyo, Japan) and 10× objective. The micro-images were taken using a MS60–2–6.3MP sCMOS camera (Nikon, Tokyo, Japan). The next set of images were taken at different time intervals (24 h and 48 h for both the cell line and supplementary, at 96 h in CAL-27 cells) until the complete wound closure of the control. In order to determine the migration rate, the images were analyzed using ImageJ software, and the percentage of the closed area was measured compared to the value obtained at T0 for each variant. The experiment was performed in triplicate.

### 2.6. Clonogenic Assay

According to Franken et al. (2007) [72] and Rafehi et al. (2011) [73], the clonogenic assay was applied to appreciate cell survival. Briefly, CAL-27 cancer cells and V79 normal cells were seeded in 12-well plates (100 cells/well) and allowed to grow for two days. After culture initiation, the 48 h treatment was applied with the specific IC_50_ values previously established for the *U. barbata* extracts and the usnic acid and 0.2% DMSO. When the treatment time expired, the growth medium was discarded and replaced with fresh complete DMEM. Until the colonies reached a minimum of 50 cells/colony (14 days), the growth medium was replaced every two days. After colonies from the control group were formed (over 50 cells/colony), the colonies were washed with PBS, fixed with ethanol, and stained with Trypan blue. Colonies were counted and analyzed with FIJI software [74]. Three independent determinations were performed.

### 2.7. Antioxidant Enzymes Activity Assay

To estimate the effect of *U. barbata* extracts on some biomarkers of oxidative stress, the activity of the main antioxidant enzymes, superoxide dismutase (SOD), catalase (CAT), and glutathione peroxidase (GPX) as well as the level of malondialdehyde (MDA), a product resulting from the lipid peroxidation, were evaluated. For these determinations, the cells were grown in DMEM, to which 10% fetal bovine serum and 1% penicillin-streptomycin were added, at 37 °C in a humidified atmosphere with 5% CO_2_ binder incubator. After obtaining the cell monolayer, the cells were treated for 6 h with IC_50_ specific values of each *U. barbata* extract. A positive control was performed, meaning 100 µM H_2_O_2_ treatment for 15 min, twice washed with cold phosphate-buffered saline (PBS) solution, and then trypsinized simultaneously with other treated groups to obtain the cell lysates. The samples were ultrasonicated in ice-cold 0.1 M potassium phosphate buffer (pH 7.4), 1.15% KCl, at a 8 × 10 cycle, power 35%, for 1 min, repeated 8 times, then centrifuged (15 min at 3000 rpm), and the supernatant was used in determinations for biomarkers of oxidative stress [75,76].

#### 2.7.1. Determination of Superoxide Dismutase Activity 

Superoxide dismutase (SOD) activity was determined according to Winterbourne’s method with slight changes, based on the SOD ability to inhibit the reduction of nitro blue tetrazolium (NBT) by superoxide radicals resulting through the reoxidation of photochemically reduced riboflavin. The degree of inhibition determined by the enzyme was estimated by measuring the treated samples and controlling extinctions at 562 nm against distilled water [77].

#### 2.7.2. Determination of Catalase Activity 

Sinha’s assay evaluates catalase (CAT) activity with minor adaptations [78]. CAT acts on hydrogen peroxide for a well-defined period, after which it is inactivated by adding a mixture of potassium dichromate-acetic acid. After CAT inactivation, the amount of unmodified oxygenated water was reduced in the acid medium, the potassium dichromate, to chromic acetate, which was determined at 570 nm. The difference between the initial and final quantity of oxygenated water in the reaction medium constitutes the amount of oxygenated water decomposed by catalase.

#### 2.7.3. Determination of Glutathione Peroxidase Activity

Glutathione peroxidase (GPx) activity was assessed using the protocol described by Fukuzawa and Tokumura [79]. GPx catalyzes the decomposition of hydrogen peroxide (H_2_O_2_) with the participation of reduced glutathione as a reducer, resulting in oxidized glutathione (G-S-S-G) and water. The remaining reduced glutathione reacts with DTNB to form a yellow complex. The intensity is measured spectrophotometrically, with the difference between the initial and final amount being directly proportional to the enzymatic activity [80].

#### 2.7.4. Determination of Malondialdehyde Levels 

Malondialdehyde (MDA) was determined by the modified method of Ohkawa et al. (1979) [76]. The principle of the method consists of the reaction of MDA, resulting from the decomposition of lipid peroxides at high temperature and in acidic medium with 2-thiobarbituric acid (TBA), forming a pink trimetine adduct MDA-TBA2, with maximum absorption at 532 nm.

The activity for each enzyme was expressed as enzyme units per mg of protein. Determination of protein concentration was realized from the same cellular lysate by the Bradford assay. The principle of the method is based on the observation that the maximum absorption for the acidic solution of Coomassie Brilliant Blue G-250 changes from 465 nm to 595 nm when protein binding occurs [81].

### 2.8. Statistical Analysis

The results of the in vitro trial were expressed as the average of three parallel tests ± standard error (ES). The difference between the mean values for each parameter was expressed by the Student’s *t*-test, using software GraphPad Prism version 8 (San Diego, CA, USA) [82].

## 3. Results

### 3.1. Preparation and Characterization of Lichen Extracts

All the data are shown in Table 1 and Appendix A. The secondary metabolite content was determined in our previous study, and the obtained results are also synthesized in Table 1 (where DE = dry extract).

**Table 1 antioxidants-10-01141-t001:** Preparation and characterization of the four *U. barbata* dry extracts.

*U. barbata* Dry Extract	Color	Temperature	Yield%	UA (mg/g)	TPC (mg/g DE)	TC (mg/g DE)
UBE	Light-brown	75–80 °C	12.52	127.21	67.3	14.7
UBM	Brown	65 °C	11.29	137.60	70.7	9.99
UBA	Yellow-brown	55–60 °C	6.36	282.78	101.09	24.4
UBEA	Brown-yellow	75–80 °C	6.27	376.73	42.40	3.85

The spectra acquisition for each of the four *U. barbata* dry extracts (Figure 1a–d) on the range 4000–400 cm^−1^ were analyzed individually to identify the main functional groups and superimposed to assess the similarity degree (Figure 1e).

The detailed FTIR spectra with recorded peaks of each extract and usnic acid are included in the Appendix A. The prominent peaks in the region 4000–1500 cm^−1^ (region of functional groups) were examined; their frequency and interpretation are synthesized in Table 2.

### 3.2. MTT Assay

In vitro screening on normal and tumor mammalian cell cultures aimed to investigate the effect of four *U. barbata* extracts on the viability of human squamous carcinoma cells CAL-27 and normal lung fibroblasts from Chinese hamster V79 using the MTT viability assay. The two cell lines were incubated with the four extracts for 24 and 48 h, with doses ranging from 25 to 200 µg/mL. Usnic acid was used as a positive control in the dose range of 1.25–10 µg/mL. The cytotoxic effect of *U. barbata* extracts on cancer CAL-27 cells is illustrated in Figure 2 and Figure 3. Thus, the methanol extract of *U. barbata* (UBA) interferes slightly with CAL-27 cell viability after 24 h treatment at the lower used dose (25 µg/mL), with a value of 77.15%, which decreased to 58.59% after 48 h of treatment. At the maximum used dose (200 µg/mL), there was a slightly more pronounced decrease in cell viability after 24 h of treatment (54.45%) and 48 h of treatment to decrease to 41.06%. The ethanol extract of *U. barbata* (UBE) behaved similarly to the methanol extract, determined at 25 µg/mL viability of 76.36% after 24 h treatment and 66.373% after 48 h treatment. 

There was a decrease in cell viability with the dose, at 200 µg/mL being recorded a value of 59.25% at 24 h and of 38.67% at 48 h. The impact of the acetone extract of *U. barbata* (UBA) upon cancer cell viability was moderate at 25 µg/mL, determining values of 73.88% at 24 h treatment and of 53.49% at 48 h. The situation changed significantly, especially at the dose of 200 µg/mL and after 48 h of treatment, when a pronounced decrease in cell viability was reported of 22.25%. The ethyl acetate extract of *U. barbata* (UBEA) determined insignificant decreases in cell viability at 24 h of 79.47% (25 µg/mL) and 60.08% (200 µg/mL). The 48-h treatment with UBEA was followed by decreases in the viability of neoplastic cells, which were not so pronounced at 25 µg/mL (66.15%), but significantly reduced at 200 µg/mL, reaching values of 33.25% (Figure 3).

In addition, the in vitro trial included testing the *U. barbata* extracts on a healthy cell line, normal lung fibroblasts from Chinese hamster, V79 (Figure 4 and Figure 5). On this cell line, it was observed that the cytotoxic effect of the four *U. barbata* extracts was generally lower compared to that manifested on the CAL-27 tumor line. Thereby, the UBM extract at 24 h of incubation exhibited a low cytotoxic effect on V79 cells, highlighting viability values of 81.83% (25 µg/mL) and 65.39% (200 µg/mL). The 48 h UBM extract treatment revealed an increase in cytotoxic effect in a dose-effect manner, the percent of living cells being 65.39% at 25 µg/mL and 30.96% at 200 µg/mL. The 24 h incubation with UBE extract induced a viability of 81.36% (25 µg/mL) and of 73.36% (200 µg/mL), respectively. The 48-h treatment with UBE was followed by a decrease in the living cell, with a percentage of 69.21% (25 µg/mL) and 30.23% (200 µg/mL). The UBA extract determined, at 24 h incubation, small decreases in cell viability, reaching 74.85% (25 µg/mL) and 61.67% (200 µg/mL) values.

Continuation of UBA treatment to 48 h led to moderate decreases in cell viability at the minimum dose (56.86%) and significant decreases at the maximum dose (25.38%). Finally, the UBEA extract, similar to the other extracts after 24 h of treatment, determined moderate cytotoxicity on V79 fibroblasts of 94.43% (25 µg/mL) and 68.48% (200 µg/mL). The cell viability showed a gradual decrease with increasing dose after 48 h of UBEA extract treatment, 60.98% at 25 µg/mL and 36.83% at 200 µg/mL (Figure 5).

In the case of both cell lines, it was found that DMSO 0.2%, in which the *U. barbata* extracts were dissolved, is non-toxic, registering values of cell viability very close to those of the untreated control. 

The cytotoxic activity of *U. barbata* extracts was better illustrated by IC_50_ values, meaning the concentration at which the studied extracts exert half of the maximal inhibitory effect. The results confirmed that the strongest antitumor effect against cancer CAL-27 cells had the UBA extract, with an IC_50_ value of 37.34 µg/mL, followed by UBEA, with an IC_50_ value of 56.78 µg/mL, UBM with IC_50_ value of 60.18 µg/mL, and finally UBE, with an IC_50_ value of 75.79 µg/mL. The average half-maximal inhibitory concentration (IC_50_) values were estimated in the case of normal V79 fibroblasts were approximately two times higher than cancer cells: UBA IC_50_ = 54.89 µg/mL, UBM IC_50_ = 123.35 µg/mL, UBE IC_50_ = 126.22 µg/mL, and UBEA IC_50_ = 140.13 µg/mL. Skehan et al. (1990) reported that NCI (US National Cancer Institute) has established three cytotoxicity groups for crude extracts obtained from natural sources: inactive extracts (IC_50_ > 100 µg/mL), moderately active (IC_50_ 20–100 µg/mL), and active (IC_50_ < 20 µg/mL) [83]. The IC_50_ values of the *U. barbata* extract on the cancer line were between 20 and 100 µg/mL; hence all the extracts were moderately active according to NCI and were about two times less active in the healthy V79 cell line, meaning a slightly differentiated cytotoxic effect between the tumor and normal cells.

### 3.3. Cell Morphology Assay

Carreño et al. (2021) highlighted that the MTT assay could only distinguish viable cells from either senescent and dead cells, but not between senescent and dead cells, because senescent cells may continue transforming MTT to formazan. They indicated that microscopic observations, like cell detachment (in adherent cell lines) and the presence of cell fragments, are essential to confirm cellular toxicity [84]. Thus, the cytotoxicity of *U. barbata* extracts was also evaluated by analysis of the cells’ shape and morphology after 24 h and 48 h treatment. As shown in Figure 6 and Figure 7, the morphological characteristics of the cells were affected both depending on the solvent used but, especially on the duration of the treatment and the concentration of the tested extracts. Hence, cell adhesion, the ability to form the monolayer, and the integrity of the cell membrane were altered. In addition, cells displayed membrane shrinkage, cell vacuolization, many cell fragments, and reduction of cellular density by the accumulation of dead cells that floated in the growth medium. 

In the normal V79 cell line, the intensity of cell response to the *U. barbata* extract application was lower than in CAL-27 cancer cells (Figure 8 and Figure 9). In both cell lines, the accumulation of the floating cells by losing adhesion ability, membrane shrinkage, and formation of abnormal cellular wrinkle occurred. These cell morphological changes led to a reduction in the cell viability but revealed a selective cytotoxic effect of the extracts. 

### 3.4. In Vitro Wound Healing Assay

Because cell migration plays a major role in the cancer metastasis process, we applied the in vitro wound healing test using the CAL-27 tumor cell line and the V79 normal cell line in order to establish the selective wound closure property for each of the four *U. barbata* dry extracts, compared to the untreated control and DMSO used as the vehicle control. For both cell lines, wound closure evaluation was performed until healing was completed in the untreated control. As shown in Figure 10, after 24 h, the wound closure rate in the CAL-27 cell line ranged from 36.82% (control) to 6.51% (UBE). The lowest healing processes were maintained after 48 h in the UBE extract treatment, the dimension of wound edges representing 75.78% of the T0 value. The complete wound closure was achieved after 96 h for both the control and 0.2% DMSO variant. At this time, the wound closure was almost complete in all extracts, being between 76.60% (UBA) and 94.43% (UBEA) compared to the control. In addition, the usnic acid was effective in promoting wound healing, reaching a 96.07% rate of wound closure. Among all the tested *U. barbata* extracts, the widest wound edges were registered for the acetone (UBA) and methanol (UBM) extract at the IC_50_ value, proving a potential inhibitory effect on metastatic progression (Figure 10 and Figure 11).

The cell progression and migration rate for wound closure was faster for the V79 cells than the CAL-27 cell line; the complete wound closure for the control and 0.2% DMSO was established at 48 h from scratch initiation. As shown in Figure 12 and Figure 13, after 24 h, the control scratch area was covered by the cells in the percentage of 87.14%, where the highest percentage of wound closure was encountered at the UBEA extract (63, 85%), and the lowest healing capacity was established after exposure to the UBA extract (38.15%). 

Prolonging the treatment time, the migration rate at 48 h after the application of *U. barbata* extracts was between 85.70% (UBA) and 96.66% (UBM). Among all *U. barbata* extracts, the highest sensitivity of the V79 cell line was also established after the UBA extract treatment, similar to the CAL-27 cell line response, but in a reduced manner. These results could prove the high tolerability and selectivity of the tested *U. barbata* dry extracts upon the normal V79 cell line.

### 3.5. Clonogenic Assay

Our studies focused on assessing whether the four *U. barbata* extracts obtained using different solvents had long-term action on survival and cell proliferation. The clonogenic assay was performed for both the CAL-27 tumor cell line and normal V79 cells treated with the IC_50_ established for each extract. As shown in Figure 14a, the clonogenic potential of the CAL-27 cells was significantly inhibited after the application of the extracts. The lowest percent of colony formation was exerted by UBA (12.24%), followed by UBEA, UBE, and UBM. The 0.2% DMSO vehicles showed similar colony development like the untreated control. In the V79 cell line, the survival potential was less than 50% only in UBE exposure (43.26%). More than 90% ability of colony formation was obtained after UBA treatment (Figure 14b). For both cell lines, standard usnic acid (UA) had no toxic effect on colony formation, with the survival fraction reaching 97.96% in the CAL-27 cells and 98.58% in the V79 cells, respectively.

### 3.6. Antioxidant Enzymes Activity Assay 

The effect of *U. barbata* extracts on some oxidative stress biomarkers such as the activity of the main antioxidant enzymes, SOD, CAT, and GPx as well as the level of (MDA) was investigated. The CAL-27 cancer cells and normal V79 cells were treated for 6 h with IC_50_ specific values of each *U. barbata* extract. A positive control was performed; the cells were treated for 15 min with 100 µM H_2_O_2_, then twice washed with PBS and trypsinized simultaneously as other treated groups to obtain the cell lysates used to estimate the mentioned oxidative stress biomarkers.

#### 3.6.1. Determination of Superoxide Dismutase Activity

Superoxide dismutase, the first line of defense of cells against ROS generated by exposure to various agents, is the principal-agent for cleaning superoxide radicals, being a powerful antioxidant that converts the toxic superoxide anion (O_2_^–^) into hydrogen peroxide and oxygen by thee dismutation reaction: 2O_2_ + 2H^+^ → H_2_O_2_ + O_2_.

The impact of different *U. barbata* extracts after 6 h-treatment with specific IC_50_ values on SOD activity in CAL-27 cells, expressed in units of SOD/mg protein, is illustrated in Table 3. SOD activity stimulation was observed by all four *U. barbata* extracts compared to the untreated control (considered 100%), with the most active stimulation recorded by UBE (212.76%), and the lowest one by UBA (180.04%). In the case of the V79 normal cells, there was an inhibition of SOD activity for all *U. barbata* extracts, the smallest value was found at UBA (40.49%), and the largest at UBM (90.44%), compared with the untreated control (100%), as seen in Table 4.

#### 3.6.2. Determination of Catalase Activity 

Catalase, an essential enzyme with the role of detoxifying ROS in all aerobic organisms, catalyzes the conversion of toxic H_2_O_2_, resulting from the dismutation reaction of SOD, in H_2_O and O_2_, in peroxisomes. SOD and CAT are complementary in their action to reduce the effects of oxidative stress. The increase in CAT activity indicates an increased capacity of this enzyme to eliminate the reactive species formed; it is an efficient purifier of H_2_O_2_ with a high affinity to the substrate.

The interference of 6-h treatment with particular IC_50_ values of *U. barbata* extracts, with the activity of CAT expressed in units of U CAT/mg protein, in CAL-27 cells, was followed by a decrease in enzymatic activity, except in the UBM extract, which registered a slight increase in CAT activity (104.43%). The lowest activity of CAT activity was registered in the UBE extract (66.36%) compared to the control (100%), as shown in Table 2. A similar trend was found in normal V79 cells. The decrease in the enzymatic activity of CAT under *U. barbata* extracts had a higher amplitude than the CAL-27 cancer cells. Thus, the most potent inhibitory action of CAT was UBEA (50.2%) and the weakest was UBA (90.96%) compared to the control (100%), as illustrated in Table 3.

#### 3.6.3. Determination of Glutathione Peroxidase Activity

Glutathione peroxidase catalyzes the decomposition of hydrogen peroxide (H_2_O_2_) with reduced glutathione as a reducing agent, resulting in oxidized glutathione (G-S-S-G) and water. The activity of GPx was expressed in U GPx/mg protein.

Treatment with *U. barbata* extracts in CAL-27 cancer cells leads to slight increases in GPx activity compared to the control group, the highest activity being found in UBM (117.7%) and the lowest in UBA (106.21%), as shown in Table 2. In addition, in normal V79 cells, stimulation of GPx activity was registered, the highest value also being found in the UBM extract (128.55%) compared with the untreated control (100%). An exception was reported for the UBA extract, which caused an inhibitory effect on GPx activity (95.9%), as rendered by Table 3.

#### 3.6.4. Determination of Malondialdehyde levels

MDA is a lipid peroxidation product, representing an important indicator of oxidative damage of macromolecules induced by ROS.

The effect of *U. barbata* extracts on redox status, especially on lipid peroxidation in the two cancer and normal cell lines, was quantified by measuring the changes in the MDA levels, as illustrated in Table 2 and Table 3. 

CAL-27 cancer cells showed a decrease in MDA levels, especially after exposure to UBA, reaching the value of 69.94% compared to the untreated control, considered as 100%. The level of MDA was the least affected by UBE compared to the control (100%), when a 91.29% value was recorded. The same trend was also observed in the case of V79 normal cells of MDA level decrease, but of different amplitudes. The most significant negative impact was determined by UBE (75.17%), and UBEA affected MDA level the least (97.24%) when compared with the control (100%). 

Our experimental results, comparable to those in the literature, through evaluation of the *U. barbata* extracts’ action on oxidative stress biomarkers, suggest their targeted interaction with the complex chain of cellular redox processes so that the increase antioxidant enzyme activity and decrease in MDA level in CAL-27 cancer cells can lead to an imbalance in antioxidant defense mechanisms, with a tendency toward prooxidants, a substrate for the cytotoxic effect, while in normal V79 cells was observed an antioxidant effect, which protects normal cells against free radicals. 

## 4. Discussion 

The phytochemical profile of *U. barbata* is already known; Salgado et al. (2018) identified a wide range of secondary metabolites in a methanol extract [85]. The metabolomics of this species belongs to different classes of chemical compounds: depsides (barbatic acid, methyl-8-hydroxy-4-O-demethylbarbatate, baeomycesic acid, 8-hydroxybarbatic acid), depsidones (connorstictic acid, fumarprotocetraric acid, hypoconstictic acid, lobaric acid), lipids (polyhydroxylated lipids), and dibenzofurans (usnic acid, placodiolic acid) On the other hand, in an acetone extract of *U. longissima*, Reddy et al. (2019) reported another dibenzofuran compound, usenamine A, with the –NH_2_ group [86]. Of all these lichen secondary metabolites, usnic acid is by far the best known and responsible for most bio-activities of *U. barbata*, and at the same time, of all lichens of the *Usnea* genus. 

The characteristics of the OH group’s absorption band at 3200 cm^−1^ can be attributed to classes of compounds such as depside, depsidones, and dibenzofurans with OH phenolic.

Placodiolic and usnic acid, being characteristic of the studied species, may be related to this absorption band, a connection strengthened by the presence of the characteristic band for the carbonyl group C=O around 1700 cm^−1^. 

All extracts were identified as single C–H bonds (2918 and 2850 cm^−1^) and double C=C bonds present in an aromatic nucleus (approximately 1629 cm^−1^), most likely belonging to aromatic acids. 

No triple-bonds were found in any extract.

The characteristic band of lactones (1736 cm^−1^) has been identified in an ethyl acetate extract, a class of compounds that frequently occurs in lichens. If their presence in the extract is confirmed, it could mean that ethyl acetate is a suitable solvent for lactone extraction. Both previously mentioned studies reported several compounds with a lactone ring: lobaric acid (δ-lactone) and both usnic and placodiolic acid (γ-lactone). Moreover, our previous study proved that the ethyl acetate extract had the highest usnic acid content [61]. 

The acetone extract shows the absorption band characteristic for N–H aromatic secondary amine at 3448 cm^−1^ as a distinctive extraction pattern. This specific band could be associated with usenamine A, as previously mentioned.

Kondratiuc et al. (2015) optimized a protein extraction method from *U. antarctica* [87]. Belonging to the same *Usnea* genus, we can correlate the existence of the corresponding C=O amide band at 1572 cm^−1^ with protein structures in the *U. barbata* acetone extract.

Finally, it could be observed that all *U. barbata* extracts showed a high grade of matching in the 4000–1500 cm^−1^ region, but with visible differences in intensities.

Diverse species of lichens fulfill numerous functions in traditional medicine, the food industry, the perfume industry, and dyes [88]. Mostly, the cell-killing effect of different lichen extracts has gained attention in the treatment of several types of cancer, as some studies have shown [48,89,90,91]. In the present study, we investigated the anticancer effect and possible mechanism of action of *U. barbata* extracts against oral cancer CAL-27 cells. The role of preferential killing cells, morphological characteristics, wound healing property, clonogenic capacity, and antioxidant enzyme activity in oral cancer after treatment with *U. barbata* extracts are discussed below.

A recent study investigated the cell-killing property of methanol extracts of *U. barbata* (MEUB) against oral cancer cells (Ca9-22, OECM-1, CAL 27, HSC3, and SCC9), showing preferential killing versus these cancer cell lines but rarely affecting normal oral cell lines (HGF-1). Ca9-22 and OECM-1 cells display the highest sensitivity to MEUB [92]. 

Other studies have revealed that different extracts of various *Usnea* species induced cell killing effects against several types of cancer cells. Thus, the methanol extract of *U. intermedia* was cytotoxic upon MCF7, MDA-MB-231, A549, and H1299 cells (IC_50_ values were 17.5, 3.0, 21.4, and 10.2 µg/mL in 72 h ATP assay) [93]. Another study found that the methanol extract of *U. filipendula* Stirt induced a cytotoxic effect against lung (A549, PC3), liver (Hep3B), and rat glioma (C6) cancer cells [94] where the IC_50_ values were 37.0, 32.9, 60.5, and 67.9 µg/mL in the 72 h ATP assay. Different extracts of *U. barbata* such as (E1) CO_2_ supercritical extract and (E2) ether fraction of Soxhlet extract, demonstrated cytotoxic action on mouse melanoma B16 and rat glioma C6 cells, their IC_50_ values being (E1) 31.21 vs. 58.20 and (E2) 43.40 vs. 69.10 µg/mL, respectively, in an acid phosphatase assay at 24 h [95].

Three extracts of lichen *U. undulata* Stirt. (*n*-hexane, acetone and methanol) were tested for cytotoxic activity on two human cancer cell lines (MCF-7, NCI-H460) using the sulforhodamine B (SRB) assay. The *n*-hexane extract manifested the highest inhibition activity, with IC_50_ 5.26 ± 0.13 µg/mL for MCF-7 and 6.83 ± 0.64 µg/mL for NCI-H460, followed by acetone extract with IC_50_ 66.44 ± 1.37 µg/mL and 92.14 ± 1.49 µg/mL, respectively. The methanol extract showed a weak cytotoxic effect against MCF-7 and NCI-H460 (IC_50_ values more than 100 µg/mL). The *n*-hexane extract demonstrated cytotoxicity toward MCF-7 cells and low damage to normal fibroblasts, which means good selectivity [96].

The WST-1 assay for cell proliferation and viability evidenced that acetone extracts of Moroccan *Evernia prunastri (E. prunastri), Ramalina farinacea (R. farinacea),* and *Pseudevernia furfuracea (P. furfuracea)* showed moderate cytotoxic effects against human prostate cancer (22RV1), human colon carcinoma (HT-29), human hepatocellular carcinoma (Hep-G2), and hamster ovarian cancer (CHO) cells lines, with IC_50_ values ranged from 42.30 to 140.24 µg/mL [93,94,95]. 

In the present study, IC_50_ values for the *U. barbata* dry extract-treated oral cancer CAL-27 cells for 48 h in the MTT assay were 37.34 (UBA), 56.78 (UBEA), 60.18 (UBM), and 75.79 (UBE) µg/mL, respectively. It can be seen that our *U. barbata* dry extracts exhibited a similar drug sensitivity to human oral cancer cells with extracts from another *Usnea* species to diverse cancer cells [95,96,97,98]. Compared with usnic acid (UA), used as a positive control, the IC_50_ value of UA in oral cancer CAL-27 was 8.34 µg/mL for the 48 h MTT assay, meaning that the studied *U. barbata* extracts contain enough usnic acid to induce antitumor activity, their lower sensitivity than that of UA being due to other compounds (polyphenols, tannins) from extracts.

While *U. barbata* extracts manifested a cytotoxic effect on the CAL-27 OSCC line, the healthy V79 cell line was about two times less active, meaning differentiated cytotoxicity between the tumor and normal cells, following other studies, where the cytotoxic activity of lichens was observed in different cancer cell lines. It has been noticed that their cytotoxic capacities in cancer cells are higher than in normal cells [53,99].

Lichens contain bioactive compounds that can be considered promising candidates for cell proliferation and migration, essential for tissue regeneration. Even if the extract of *U. articulata* is empirically administered to treat wounds and bruises [100], the in vitro studies regarding the wound healing ability are few and have been conducted in the last decade. Thus, Yang et al. (2016) mention an inhibition of migration invasion of A549 cell line after treatment with *U. florida* (5 μg/mL) acetone extract, the activity being due to some secondary metabolites such as depside, depsidone, dibenzofurans, and depsone [51]. On the other hand, the usnic acid derivates like sodium usnic acid may promote skin wound healing by increasing fibroblast proliferation [101]. Burlando et al. (2009) showed that the dibenzofuran derivative (+)-usnic acid polyketides compound (1) expressed wound healing at low doses against some malignant cell lines (MM98, A431) [102]. These results proved that this compound may be useful in the prevention of hyperproliferative syndromes. Furthermore, some usnic acid enamines (compounds 2–11) demonstrated, at subtoxic doses, the in vitro and in vivo wound closure of the keratinocytes, making them useful in healing-promoting and obtaining anti-aging skin formulations [103]. Our results are similar to the mentioned scientific literature: the four *U. barbata* extracts induced high toxicity in the CAL-27 oral cancer cell line, as the wound healing activity occurred slowly, and after 96 h of the treatment, the remaining wound area was between 23.40% (UBA), followed by UBM (21.08%), and UBE (11.76%). The intense effect on wound closure was established with UBEA treatment, and the wound area had a close value of 5.57% to the UA exposure (3.3%).

The in vitro cell survival test assesses the ability of single cells to survive and proliferate to form colonies used to study the effects of some natural or synthetic compounds and also of radiation impact [72,104]. Our selectivity response between tumor and normal cells was similar to that of [105], who mentioned that atranorin (ATR), the common lichen secondary metabolites of *Parmeliaceae*, reduced the clonogenic potential of mouse breast cancer (4T1) cells compared to normal mammal non-malignant epithelial (NMuMG) cells. Guzow-Krzemińska et al. (2019) also tested some usnic acid derivates with high bioavailability and selectivity against cancer cells [106]. Thus, the UA compound has an inhibitory potential of clonogenic and cell survival on some breast cancer cells, but it is safe for non-tumorigenic MCF-10A mammary epithelial cells. Some syntheses of usnic acid derivatives have proven the significant inhibition of the clonogenic potential of MCF-7, HeLa, and PC-3 cancer cells [107].

Antioxidant enzymes such as glutathione peroxidases (GPx), catalases (CAT), and superoxide dismutases (SOD) catalyze the intracellular reduction of reactive oxygen species (ROS). Nevertheless, sometimes, antioxidant defense mechanisms are not enough to assure a redox balance that tilts toward prooxidants. The body develops oxidative stress, thus causing toxicity and genome impairment [108].

Lichens prevent mutagenesis and carcinogenesis through the inhibition of cellular macromolecule oxidation [109]. Therefore, the protective effects of lichens against oxidative impairment can be determined by monitoring the above-mentioned oxidative stress markers [110].

One of the most widely studied lichen compounds in pharmacological effects including its antioxidant potential is usnic acid (UA). UA determined increased levels of superoxide dismutase (SOD), glutathione peroxidase (GPx), and reduced glutathione (GSH) and reduced lipid peroxidation in gastric ulcer in rats [111]. Another study demonstrated that usnic acid protects astrocytes against hydrogen peroxide-induced oxidative stress by increasing cell viability and inhibiting intracellular ROS production; this action seems partially related to its peroxyl scavenger ability [112]. Usnic acid was also effective as an antioxidant agent against lipopolysaccharide-induced lung injury by decreasing the hydrogen peroxide, myeloperoxidase, and malondialdehyde levels and increasing the superoxide dismutase (SOD) and reduced glutathione (GSH) levels [113]. 

Against this background, our results regarding the effect of *U. barbata* extract on markers of oxidative stress are similar to those reported in other studies. Thus, UBA acetone and UBEA ethyl acetate extracts, which possessed the highest content of usnic acid (282.78 and 376.73 mg UA/g *U. barbata* extract, respectively) [61] and showed the most substantial cytotoxic effect on CAL-27 cancer cells, developed a prooxidant behavior, causing oxidative stress, and therefore a cytotoxic effect. Instead, in normal V79 cells, it induced an antioxidant effect, which protects normal cells against free radicals. 

Finally, these four extracts of *U. barbata* in different solvents (methanol, ethanol, acetone, and ethyl acetate) could represent possible antitumor agents against cancer CAL-27 cells in a concentration- and time-dependent manner. Based on the MTT assay and IC_50_ values, the most potent anticancer property was shown by acetone and ethyl acetate extracts, which had the highest content of usnic acid, responsible for the anticancer effect. The same extracts induced a prooxidant behavior in CAL-27 cancer cells, causing oxidative stress and thus a cytotoxic effect. In normal V79 cells, an antioxidant effect was determined, which protects normal cells against free radicals. All *U. barbata* extracts induced morphological modifications (losing adhesion property, membrane shrinkage, formation of abnormal cellular, vacuolization) of different amplitudes, correlated with the tested extract, the applied dose, and the cell line. The in vitro wound-healing assay revealed that the most powerful effect on wound closure was established after ethyl acetate extract exposure. The clonogenic test showed that certain extracts of *U. barbata* attenuated the ability of cancer cells to form colonies, thereby reducing the ability to form tumors.

## 5. Conclusions

The novelty of our study consists in the evaluation anticancer properties evaluation of four autochthonous *U. barbata* dry extracts in various solvents (methanol, ethanol, acetone, and ethyl acetate) and comparative analysis of the obtained results. Furthermore, their activities were examined on the CAL-27 tongue squamous carcinoma cell line and V79 normal cells, noting the differentiated cytotoxicity, higher in tumor cells than in normal ones. Another significant aspect supporting this statement is that our tests such as clonogenic, wound healing, and antioxidant enzymes activity assays regarding the *U. barbata* studied extracts, have rarely appeared in the previous research concerning the biological effects of lichens. In addition, relatively few antitumor studies have been performed on this OSCC cell line to analyze the previously mentioned processes.

The obtained data could enrich the existing information in the scientific databases, which must be constantly updated by in vitro screening of the bioactivities of *U. barbata* extracts as well as the usnic acid impact on both cell lines used in our study. Our results suggest that *U. barbata* dry extracts may exhibit anticancer pharmacological efficacy, providing multiple and specific targeting effects through a variety of secondary metabolites that they contain and could generate a synergism, a valuable property in cancer therapy. 

Further research could explore the effect of *U. barbata* extracts against other cancer cell lines, also analyzing other crude extract constituents and evaluating their anticancer property. Moreover, deepening investigations regarding other mechanisms of action that express the cytotoxic effect in cancer cells could be helpful in considering the lichens from Romania as potential sources of antitumor drugs.

## Figures and Tables

**Figure 1 antioxidants-10-01141-f001:**
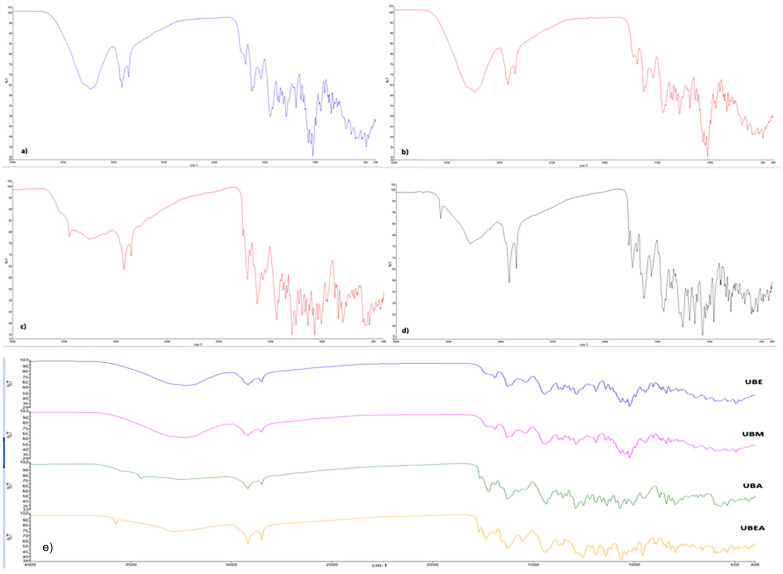
FTIR spectra of *U. barbata* extracts. (**a**) in ethanol (UBE); (**b**) in methanol (UBM); (**c**) in acetone (UBA); (**d**) in ethyl acetate (UBEA); (**e**) Overlapping FTIR-ATR spectra of all extracts: Blue—extract in ethanol, Purple—extract in methanol, Green—extract in acetone, Orange—extract in ethyl acetate.

**Figure 2 antioxidants-10-01141-f002:**
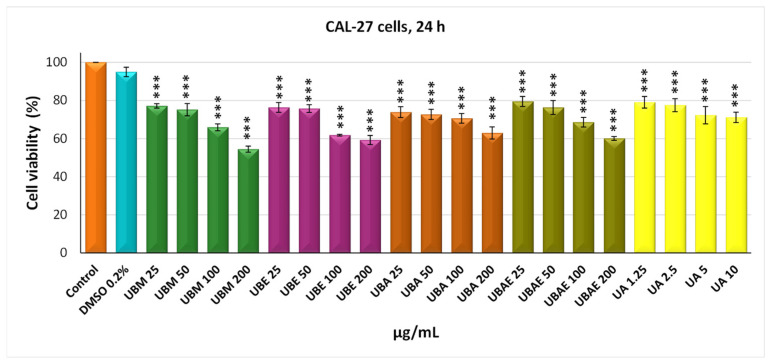
The viability of the CAL-27 cells after *U. barbata* extracts (µg/mL) treatment (24 h). The results represent the mean ± SE of three independent experiments (*** *p* < 0.001) when comparing the untreated control (*t*-test).

**Figure 3 antioxidants-10-01141-f003:**
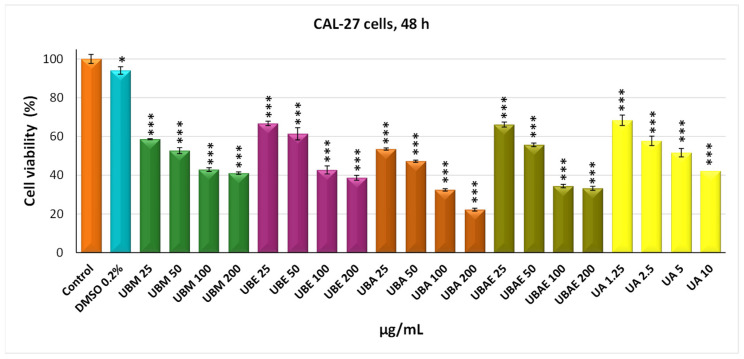
The viability of the CAL-27 cells after treatment of the *U. barbata* extracts (µg/mL) (48 h). The results represent the mean ± SE of three independent experiments (* *p* < 0.05, *** *p* < 0.001) when comparing the untreated control (*t*-test).

**Figure 4 antioxidants-10-01141-f004:**
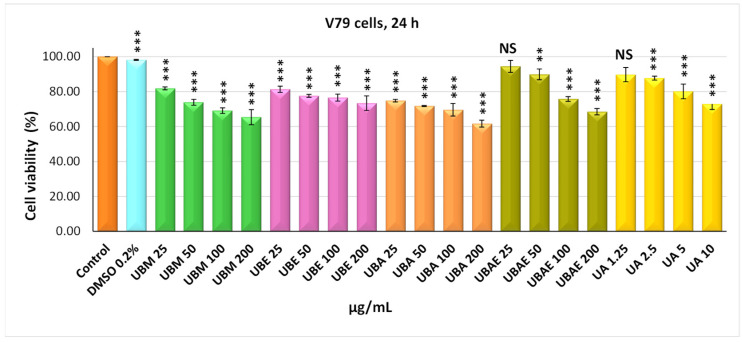
The viability of the V79 cells after treatment of *U. barbata* extracts (µg/mL) (24 h). The results represent the mean ± SE of three independent experiments (** *p* < 0.01, *** *p* < 0.001) when compared to the untreated control (*t*-test).

**Figure 5 antioxidants-10-01141-f005:**
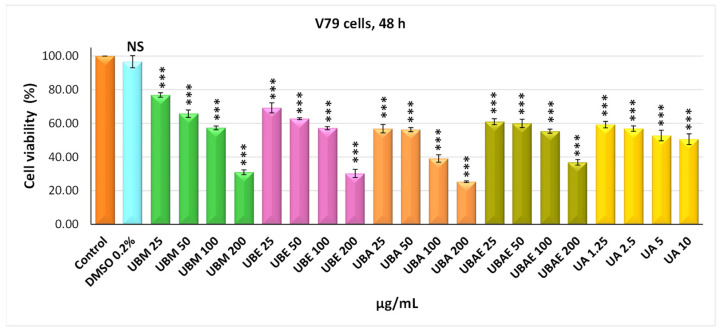
The viability of the V79 cells after *U. barbata* extracts (µg/mL) treatment (48 h). The results represent the mean ± SE of three independent experiments (*** *p* < 0.001) when compared to the untreated control (*t*-test).

**Figure 6 antioxidants-10-01141-f006:**
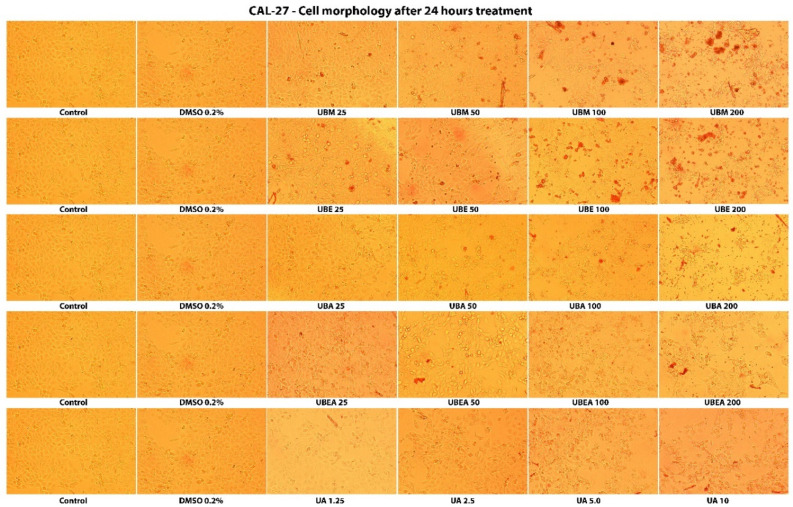
The morphology of the tumor CAL-27 cells after 24 h contact of the *U. barbata* extracts. The images were captured using an inverted microscope and 10× objective.

**Figure 7 antioxidants-10-01141-f007:**
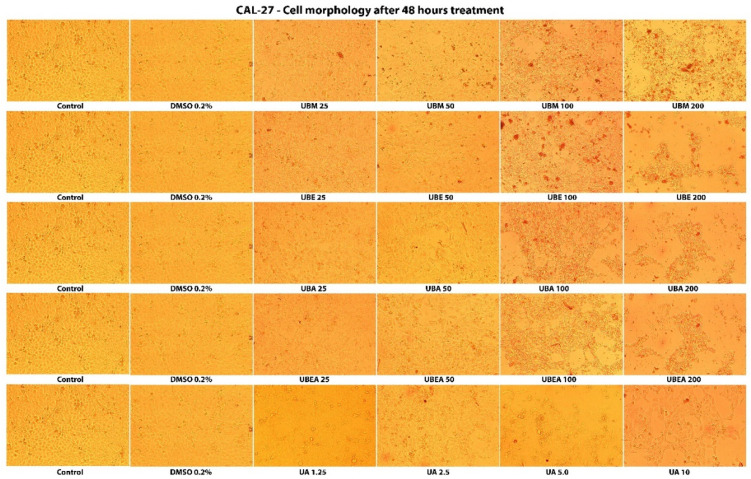
The morphology of the tumor CAL-27 cells after 48 h contact of the *U. barbata* extracts. The images were captured using an inverted microscope and 10× objective.

**Figure 8 antioxidants-10-01141-f008:**
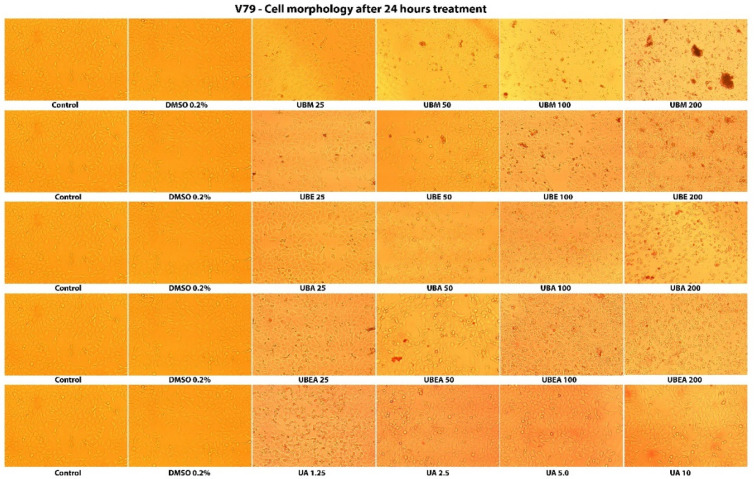
The morphology of the normal V79 cells after 24 h contact of the *U. barbata* extracts. The images were captured using an inverted microscope and 10× objective.

**Figure 9 antioxidants-10-01141-f009:**
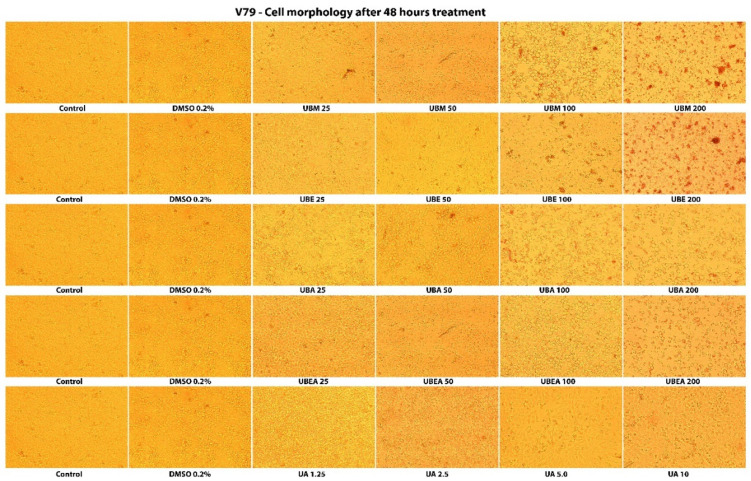
The morphology of the normal V79 cells after 48 h contact of the *U. barbata* extracts. The images were captured using an inverted microscope and 10× objective.

**Figure 10 antioxidants-10-01141-f010:**
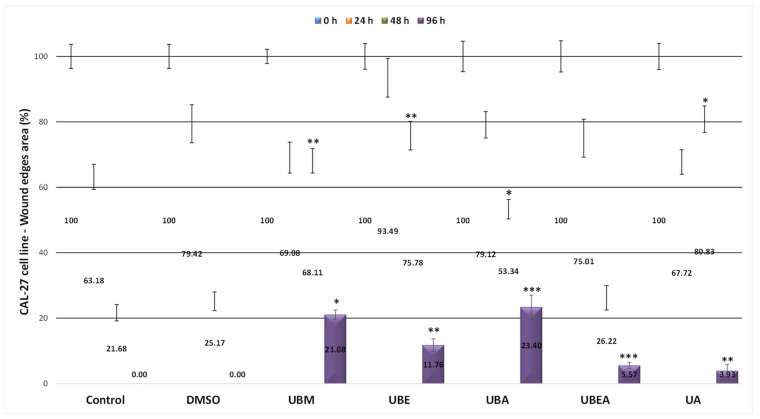
The effect of *U. barbata* extracts on the tumor CAL-27 cell migration process assessed by the in vitro scratch wound assay. The results represent the means ± SD of three independent experiments. The results represent the mean ± SE of three independent experiments (* *p* < 0.05, ** *p* < 0.01, *** *p* < 0.001) when comparing the untreated control (*t*-test).

**Figure 11 antioxidants-10-01141-f011:**
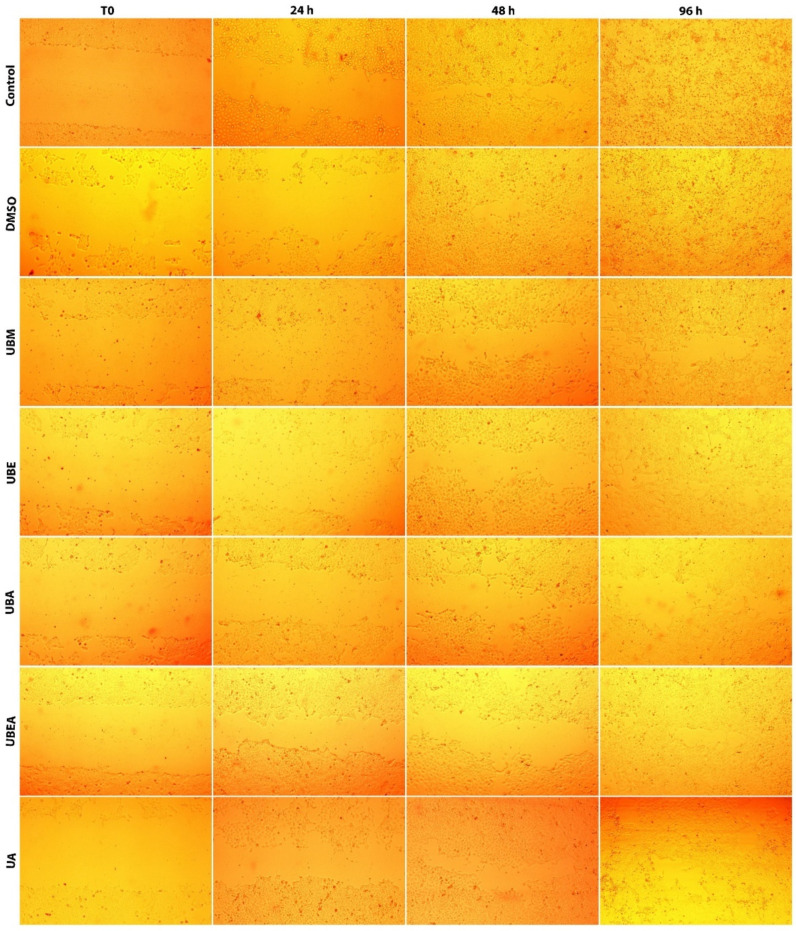
Microscopic images representing the in vitro wound healing process of tumor CAL-27 cells after treatment of *U. barbata* extracts using IC_50_ doses.

**Figure 12 antioxidants-10-01141-f012:**
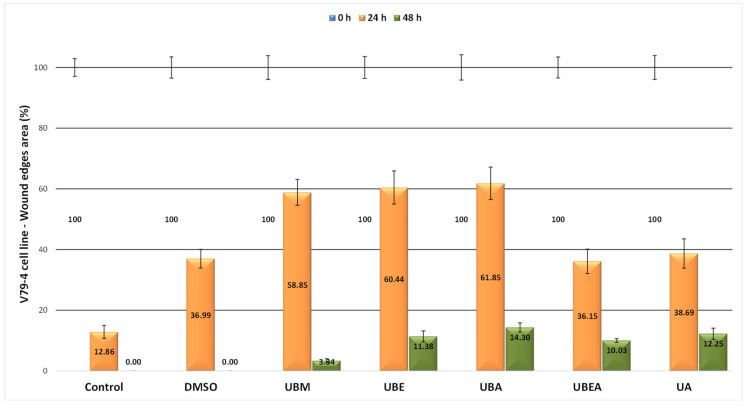
The effect of *U. barbata* extracts on the normal V79 cell migration process assessed by the in vitro scratch wound assay. The results represent the means ± SD of three independent experiments.

**Figure 13 antioxidants-10-01141-f013:**
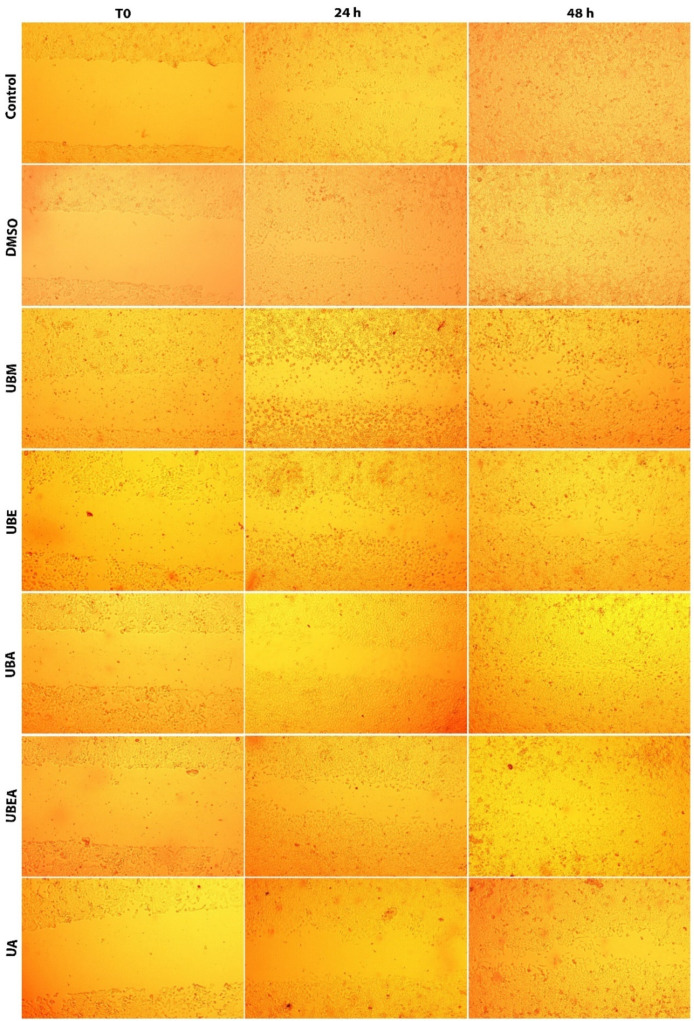
Microscopic images represent the in vitro wound healing process of normal V79 cells after treatment of *U. barbata* extracts using IC_50_ doses.

**Figure 14 antioxidants-10-01141-f014:**
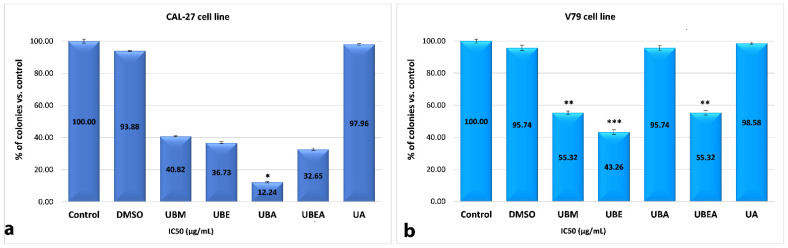
Effect of *U. barbata* extracts on the colony formation of cancer CAL-27 cell line (**a**) and normal V79 cells (**b**) cultivated in 24-well plates in triplicate. All data represent the mean ± SE compared to the control. ** *p* < 0.0, *** *p* < 0.001, *t*-test.

**Table 2 antioxidants-10-01141-t002:** IR spectrum analysis of the four extracts in the range of 4000–1500 cm^−1^.

*U. barbata* Extract	UBA		UBE		UBM		UBEA	
Domain (cm^−1^)	Frequency	Interpretation	Frequency	Interpretation	Frequency	Interpretation	Frequency	Interpretation
4000–2500 cm^−1^ single bonds region	3448 (sharp)	N–H secondary amine in aromatic nucleus	NA		NA		3575 (sharp)	OH (phenol)
	3242 (broad)	OH (phenol)	3230 (broad)	OH (phenol)	3235 (broad)	OH (phenol)	3289 (broad)	OH (phenol)
	2918	C–H (alkane) stretching	2918	C–H (alkane) stretching	2918	C–H (alkane) stretching	2918	C–H (alkane) stretching
	2850	C–H stretching	2850	C–H stretching	2851	C–H stretching	2849	C–H stretching
2500–2000 cm^−1^ triple bonds region	NA		NA		NA		NA	
2000–1500 cm^−1^ double bonds region	1767	C=O stretching	NA		NA		1771	C=O stretching
	1721	C=O (carboxylic acid)	NA		NA		1736	(δ-lactone)
	1691	C=O	1691	C=O	1692	C=O	1693	C=O
	1627	Aromatic compound C=C	1629	Aromatic compound C=C	1629	Aromatic compound C=C	1628	Aromatic compound C=C
	1572	C=O (amides)	NA		NA		NA	
	1542	Carboxylic OH	1537	Carboxylic OH	1541	Carboxylic OH	1557	Carboxylic OH

**Table 3 antioxidants-10-01141-t003:** The activity of SOD, CAT, GPx, and MDA levels in CAL-27 cells after IC_50_ *U. barbata* extract treatment.

	U SOD/mg Protein	U CAT/mg Protein	U GPx/mg Protein	MDA (mM/mg Protein)
Control	0.432 ± 0.065	3.452 ± 0.058	0.00116 ± 0.0001	11.694 ± 0.309
DMSO 0.2%	0.803 ± 0.121 *	3.115 ± 0.143	0.00116 ± 0.0000	8.194 ± 0.599 **
UBM	0.788 ± 0.124	3.605 ± 0.093	0.00136 ± 0.0000 *	10.302 ± 1.758
UBE	0.917 ± 0.133 *	2.292 ± 0.095 ***	0.00134 ± 0.0000 *	10.674 ± 2.163
UBA	0.777 ± 0.217	3.119 ± 0.103 *	0.00123 ± 0.0000	8.180 ± 0.602 **
UBEA	0.807 ± 0.153	3.110 ± 0.225	0.00133 ± 0.0000 *	9.518 ± 1.377
UA	0.890 ± 264	3.395 ± 0.216	0.00144 ± 0.0000	10.205 ± 0.519
H_2_O_2_ 100 µM	1.307 ± 0.04 ***	3.897 ± 0.269	0.00146 ± 0.0001 **	9.159 ± 0.532 **

The results represent the mean ± SE of the three independent experiments compared to the effects obtained in the untreated control (*t*-test); * *p* < 0.05; ** *p* < 0.01; *** *p* < 0.001.

**Table 4 antioxidants-10-01141-t004:** The activity of SOD, CAT, GPx, and MDA levels in the V79 cells after IC_50_ *U. barbata* extract treatment.

	U SOD/mg Protein	U CAT/mg Protein	U GPx/mg Protein	MDA (mM/mg Protein)
Control	0.969 ± 0.081	2.9547 ± 0.035	0.0015 ± 0.0000	10.150 ± 0.492
DMSO 0.2%	0.774 ± 0.036	2.1940 ± 0.305 **	0.0017 ± 0.0001	9.177 ± 0.837
UBM	0.876 ± 0.071	1.4983 ± 0.059 ***	0.0020 ± 0.0000 ***	8.802 ± 0.378
UBE	0.609 ± 0.050 **	1.9495 ± 0.102 ***	0.0018 ± 0.0000 **	7.631 ± 0.065 **
UBA	0.393 ± 0.060 *	2.6878 ± 0.185	0.0015 ± 0.0001	9.555 ± 0.221
UBEA	0.872 ± 0.108	1.4831 ± 0.270 ***	0.0017 ± 0.0000	9.871 ± 0.882
UA	0.555 ± 0.110 *	0.4497 ± 0.242 ***	0.0017 ± 0.0001	7.917 ± 0.240 *
H_2_O_2_ 100 µM	1.017 ± 0.124	3.2741 ± 0.341	0.0020 ± 0.0001 **	9.016 ± 0.876

The results represent the mean ± SE of the three independent experiments compared to the effects obtained in the untreated control (*t*-test); * *p* < 0.05; ** *p* < 0.01; *** *p* < 0.001.

## Data Availability

Data are contained within the article and supplementary materials.

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
