# Peer review of "In Vitro Anticancer Activity and Oxidative Stress Biomarkers Status Determined by Usnea barbata (L.) F.H. Wigg. Dry Extracts"

_antioxidants, 2021, doi:10.3390/antiox10071141_

Round 1

Reviewer 1 Report

The current work is interesting as it documents the potential antineoplastic activity of 4 different extracts of a lichen from Romania. The work tries to investigate the molecular mechanisms undelying its activity. I have some comments and some big concerns for the authors.

  • typology and origin of CAL-27 and V79 should be reported also in the abstract
  • line 61, "Natural compounds represent a rich source of bioactive molecules" shoucl sound "Natural PRODUCTS represent rich sourceS of bioactive molecules"
  • Due to several minor mistakes in grammar and orthography that I noticed in the text (for instance in the sentence reported in the previous point), I strongly suggest a revision of the English version of the MS. In addition, line 73 an unuseful bracket is present; line 74 the full stop is necessary; line 84 a space is required between the following two terms "[27]and"; etc. Please, check carefully the whole text. 
  • More details about the sampling of the lichen is required. How can the authors be sure of the genetic identity of the sampled material? The exact volumes and weights of solvents and lichens and the extraction procedures must be reported
  • MTT assay is able to measure cell growth and not cell toxicity. Indeed, the incapacity of cells to proliferate can be associated to also to other phenomena than cell death, such as cell cycle arrest or differentiation. I suggest to perform a specific test to minitor cell death, for instance the trypan blue-esclusion test
  • What is the necessity to dilute U. barbata extracts in DMSO? The molecules present in the extracts are already solubilized... the further dissolution in DMSO could compromise the resuspension of some polar metabolites. This choice is not consistent. Please, explain this issue.
  • The authors declare "The untreated cells with the same volume of culture medium".. it is a big problem. Indeed, CNT cells should be treated with the vehicle, that is DMSO or the other solvents. This would mean that all controls are not correct.
  • Moreover, continuing the previous point, even if the authors used DMSO as control, it is not totally correct because in each extract, beyond the lichen metabolites, also the extraction solvent used in the 4 procedures is present and it is known in literature that ethyl acetate, acetone, ethanol can be slighly toxic for cells.. A specific control for each extract should have been carried out. the present procedure is not correct and all values should have been compared to the relative control.
  • line 137, the exact concentrations should be reported
  • the paragraph 3.1. include the word "Preparation" related to the extracts but no information about that is mentioned.
  • Figure 1a,b,c and d needs of units of measure on the axis and the value on the y-axis
  • Table 1 is very confusing and should be position in vertical and not in orizontal form
  • which software was used to perform  Student's t-test?
  • Observing MTT assay at 48 h, it seems that the extracts are very toxic also on normal cells (even if the authors declare the contrary, line 322-324). This fact would mean that lichen extracts could determine side effects on normal healthy tissues, representing non adequate anticancer products.
  • Fig 12, no significant data are present here?
  • the chemical characterization of the extract is not clear.. FTIR did not provide information about the composition of the phytocomplexes that the authors extracted from the lichens.. an HPLC-DAD and/or LC-MS analysis is required, at least to identify the major components of the extracts.
  • in the introduction the authors should better describe the great antineoplastic activity of natural extracts, acting according to several molecular mechanisms, such as apoptosis, necroptosis, senescence, differentiation, cell cycle arrest, etc. They should mention some works reporting these different types of effect of natural compounds on human cells, for instance:  Bioorganic Chemistry, 2020, 104: 104317; Phytomedicine, 2018, 46: 1-10; Journal of agricultural and food chemistry, 2005, 53.5: 1776-1781; Experimental cell research, 2011, 317.1: 82-93.

Reviewer 2 Report

Reviewer's comment on Manuscript Number: antioxidants-1297466

The manuscript entitled “In vitro Anticancer Activity and Oxidative Stress Biomarkers 2 Status Determined by Usnea barbata (L.) F.H. Wigg. Dry Extracts ” falls within the scope of Antioxidants.

The manuscript is certainly very interesting and valuable. This study could stimulate more research to further the exploration of the problem.  However, I have a few remarks:

  1. First of all, the English language must be checked and improved throughout the text
  1. Why Authors have chosen particular solvents (methanol, ethanol, acetone, and ethyl acetate)? There is no explanation.
  2. Why composition of particular extracts was not evaluated? It would give information about particular compounds that can be responsible for the effects.
  3. Please emphasize what is the originality of your article and what gap it comes to fill in the existing literature?

I propose to accept this paper for publication in Antioxidants after major amendments.

Round 2

Reviewer 1 Report

The paper has been improved as requested. It can be published now.

Reviewer 2 Report

The manuscript was significantly improved and can be accepted in present form.